# Effects of Repetitive Transcranial Magnetic Stimulation over Prefrontal Cortex on Attention in Psychiatric Disorders: A Systematic Review

**DOI:** 10.3390/jcm8040416

**Published:** 2019-03-27

**Authors:** Larissa Hauer, Johann Sellner, Francesco Brigo, Eugen Trinka, Luca Sebastianelli, Leopold Saltuari, Viviana Versace, Yvonne Höller, Raffaele Nardone

**Affiliations:** 1Department of Psychiatry, Psychotherapy and Psychosomatic medicine, Christian Doppler Medical Center, 5020 Salzburg, Austria; l.hauerl@salk.at; 2Department of Neurology, Christian Doppler Klinik, Paracelsus Medical University, 5020 Salzburg, Austria; j.sellner@salk.at (J.S.); e.trinka@salk.at (E.T.); 3Department of Neurology, Franz Tappeiner Hospital, 39012 Merano, Italy; dr.francescobrigo@gmail.com; 4Department of Neurosciences, Biomedicine and Movement Sciences, University of Verona, 37100 Verona, Italy; 5Centre for Cognitive Neurosciences Salzburg, 5020 Salzburg, Austria; 6University for Medical Informatics and Health Technology, UMIT, 6060 Hall in Tirol, Austria; 7Department of Neurorehabilitation, Hospital of Vipiteno, 39049 Vipiteno, Italy; luca.sebastianelli@sabes.it (L.S.); viviana.versace@sabes.it (V.V.); 8Department of Neurology, Hochzirl Hospital, 6170 Zirl, Austria; leopold.saltuari@tilak.at; 9Department of Psychology, University of Akureyri, 6000 Akureyri, Iceland; yvonne@unak.is

**Keywords:** repetitive transcranial magnetic stimulation, attention, dorsolateral prefrontal cortex, depression, schizophrenia, autism

## Abstract

Repetitive transcranial magnetic stimulation (rTMS) may be effective for enhancing cognitive functioning. In this review, we aimed to systematically evaluate the effects of rTMS on attention in psychiatric diseases. In particular, we searched PubMed and Embase to examine the effectiveness of rTMS administered to the dorsolateral prefrontal cortex (DLPFC) on this specific cognitive domain. The search identified 24 articles, 21 of which met inclusion and exclusion criteria. Among them, nine were conducted in patients with depression, four in patients with schizophrenia, three in patients with autism spectrum disorder (ASD), two in patients with attention deficit hyperactivity disorder, one each in patients with Alzheimer’s disease and in patients with alcohol or methamphetamine addiction. No evidence for cognitive adverse effects was found in all the included rTMS studies. Several studies showed a significant improvement of attentional function in patients with depression and schizophrenia. The beneficial effects on attention and other executive functions suggest that rTMS has the potential to target core features of ASD. rTMS may influence the attentional networks in alcohol-dependent and other addicted patients. We also reviewed and discussed the studies assessing the effects of rTMS on attention in the healthy population. This review suggests that prefrontal rTMS could exert procognitive effects on attention in patients with many psychiatric disorders.

## 1. Introduction

Attention is a cognitive and behavioral process that selectively focuses on individual aspects of subjective or objective information, allowing through voluntary top-down and automatic bottom-up mechanisms to selectively process or inhibit contents from the multiplicity of sensory inputs over different domains [1,2,3]. Attention facilitates or impairs other cognitive functions, such as memory, language, problem solving, and reflects complex interactions of multiple independent systems distributed within the brain [4,5].

Psychiatric disorders can also lead to attention deficits. Dysfunctions in attentional processes and selective set-shifting have been reported in depressed individuals [6]. Schizophrenia presents with positive clinical features but also with negative clinical features, such as attentional deficits [7]. In adult patients with attention deficit hyperactivity disorder (ADHD), cognitive disturbances are more pronounced than in the pediatric population [8] and are most evident as deficiencies of executive functions and attention [9,10]. In autism, the selective attention has been shown to be impaired even in situations where behavior is normal; especially a deficit in rapid attention shifting has been observed in behavioral tasks shifting between sensory modalities, spatial locations, and object features [11,12,13].

Attention does not localize anatomically [14] and is therefore difficult to study. However, frontal regions are particularly active during tasks of alerting attention [5]. Indeed, neuroimaging studies have demonstrated the engagement of the left dorsolateral prefrontal cortex (DLPFC) in executive functioning, and more specifically during selective attention. In particular, a functional magnetic resonance imaging (fMRI) study indicates the posterior DLPFC was active during a bimodal divided attention condition [15]. The posterior DLPFC may support the increased working memory load associated with divided, compared to selective attention.

If delivered repetitively, transcranial magnetic stimulation (TMS) can influence brain function and induce changes in neuroplasticity, also in brain regions recruited by attentional processes. Indeed, repetitive TMS (rTMS) can modulate cortical excitability, inducing lasting effects [16]. Therefore, rTMS has evolved into a powerful neuroscientific tool allowing to interfere transiently with specific brain functions.

A number of rTMS studies which targeted the DLPFC have shown significant improvements in cognitive function scores using both short- and long-term stimulation paradigms [17,18,19,20,21]. It might be of interest to explore whether rTMS could serve as an intervention in disorders with attention deficits. A number of studies has specifically targeted attention, while many others assessed broader effects.

The aim of this review was to summarize the most specific studies assessing the effects of rTMS over DLPFC on attentional processes in subjects with psychiatric disorders.

## 2. Transcranial Magnetic Stimulation

rTMS is a noninvasive and safe brain stimulation technique that uses brief, intense pulses of electric current delivered to a coil placed on the subject’s head in order to generate an electric field in the brain via electromagnetic induction. rTMS has been proven to influence cortical excitability and the metabolic activity of neurons. Indeed, the induced electrical field modulates the neural transmembrane potentials and, thereby, neural activity. These effects depend on the intensity, frequency, and number of pulses applied, the duration of the course, the coil location and the type of coil used. RTMS can be applied as continuous trains of low-frequency (LF, 1 Hz) or bursts of higher frequency (HF, ≥5 Hz) rTMS. In general, LF rTMS is thought to reduce, and HF rTMS is thought to enhance excitability in the targeted cortical region [22,23,24]. The physiological impact of rTMS and other neuromodulatory techniques involves synaptic plasticity, specifically long-term potentiation and long-term depression.

However, standard coils used in research and the clinic for rTMS are not capable of directly stimulating deep brain regions. The Heased coil (H-coil) is likely to have the ability of deep brain stimulation without the need of increasing the intensity to extreme levels [25]. Deep TMS (dTMS) thus enables deeper noninvasive cortical stimulation at an effective depth of approximately 3 cm depending on the coil’s design and the stimulation intensity.

There is a sufficient body of evidence to accept with level of recommendation A (definite efficacy, Evidence Based Health Care) the analgesic effect of HF rTMS applied over the primary motor cortex contralateral to pain and the antidepressant effect of HF rTMS applied over the DLPFC [24]. Overall, rTMS techniques have been shown to have potential therapeutic efficacy in cognitive neuroscience [26]. In turn, these techniques have attracted worldwide attention as possible therapeutic tools for various neurological and psychiatric conditions [24,27].

## 3. Material and Methods

In order to identify relevant articles for this review, we searched the MEDLINE, accessed by PubMed (1966–August 2018) and EMBASE (1980–August 2018) electronic databases were searched using the medical subject headings (MeSH) and free terms: “repetitive transcranial magnetic stimulation” OR “rTMS” AND “attention” OR “attentional” OR “attentive” AND “dorsolateral prefrontal cortex” OR “DLPFC”. Only original research articles were considered eligible for inclusion. Review articles or single case reports were excluded. The search was limited to studies written in English. Studies that met the following criteria were included: rTMS was conducted to patients with psychiatric diseases or neurological disorders with behavioral symptoms; administration site of rTMS was the DLPFC; the effect of rTMS on the cognitive domain attention was examined. In contrast, rTMS studies with animals as well as studies in which rTMS stimulation was administered on sites other than the DLPFC were excluded. Moreover, we included only studies that focused exclusively on attention, while studies with a broader scope within the umbrella concept of executive functions were excluded.

Full-text articles were retrieved for the selected titles, and reference lists of the retrieved articles were searched for additional publications. When data was missing or incomplete, principal investigators of included trials were contacted and additional information was requested. The titles and abstracts of the initially identified studies were screened by two authors to determine whether they satisfied the selection criteria. The methodological quality of each study and risk of bias were independently assessed, focusing on blinding, and any disagreement was solved through discussion. This search strategy yielded 24 results, three of which were excluded after reading the full paper, thereby leaving 21 studies which contributed to this review.

A flow-chart (Figure 1) shows the selection/inclusion process.

## 4. Results

The demographic characteristics of the patients in all included articles are shown in Table 1.

The description of the rTMS interventions in the reviewed articles is shown in Table 2.

### Healthy Individuals

The breakdown of specific brain areas or neurotransmitter systems leads to selective disruptions of attentional networks in both healthy aging and pathological conditions [28]. The neural mechanisms underlying the ability to divide attention between multiple sensory modalities are still poorly understood [29].

The reviewed studies contribute to the understanding of the relationship between the DLPFC and attentional control, and suggest possible therapeutic applications for HF or LF rTMS.

These findings are consistent with those from several experimental studies in healthy humans.

Both single tasks demanding focused attention and dual task conditions requiring divided attention activate a widespread, mainly right-sided network including dorsolateral and ventrolateral prefrontal structures, superior and inferior parietal cortex, and anterior cingulate gyrus [30]. Vohn et al. performed fMRI in healthy subjects who underwent two within-modality (auditory/auditory, visual/visual) and one cross-modality (auditory/visual) divided attention task, as well as related selective attention control conditions [34]. The authors reported a significant activation in a predominantly right hemisphere network involving the PFC, the inferior parietal cortex, and the claustrum. Healthy subjects recognized fewer items after TMS over the left DLPFC than over the right DLPFC during encoding under full attention, while they produced fewer items after TMS over the right DLPFC in encoding under divided attention compared to a sham condition [31]. Taken together, these results favor the view that the right DLPFC is of special importance for attention, except for the last study which would point to a higher relevance of the left, compared to the right DLPFC.

It should be considered that selective and divided bimodal attention are concepts based on distinct neural processes. In fact, selective attention involves modulation of activity in the sensory cortices, while divided attention is achieved for most individuals via recruitment of the DLPFC [49].

TMS over PFC induced a significant reduction of performance time for both the verbal and visuo-spatial tests, thus suggesting the importance of this area in performing tasks requiring a high level of controlled attention [32].

Furthermore, 5 Hz rTMS over right DLPFC exerts remote effects on the activity of areas that functionally interact with this area during attentional processes [33].

HF rTMS over the right DLPFC was suggested to have an effect on top-down attentional processes by modulating the attentional set [35]. This is of interest, since top-down modulation mediated by the prefrontal cortex is a causal link between early attentional processes and subsequent memory performance [36].

Divided attention performance was significantly impaired about 30–60 min after a single rTMS session was applied over the left DLPFC, compared to a sham condition one week apart [26].

Daily HF-rTMS can improve attentional control in normally aging individuals [37]. Subjects who received five daily stimulation sessions of 10 Hz HF rTMS over the left DLPFC showed improved performance in reaction time during incongruent trials (i.e., those with distracting information) after HF-rTMS treatment compared with pretreatment assessment.

## 5. Results

### 5.1. Depression

Several studies assessed cognitive performance effects in patients with depression receiving rTMS. No major changes in the Continous Performace Task assessing attention and in other cognitive tests were observed in the first study of Speer and colleagues after LF or HF rTMS administered over the DLPFC [40]. Later studies assessed attention using psychometric tests, such as the d2 test, the sustained attention in the Cambridge Neuropsychological Test Automated Battery (CANTAB), the Test of Attentional Performance, and failed to find any significant effects of either HL or LF rTMS applied over the DLPFC [38,39,44]. Only one study using H-coils demonstrated that unilateral prefrontal left stimulation with H1/HIL-coils significantly improved the score on the rapid visual processing test as measured with the CANTAB [45].

A systematic review and meta-analysis of outcomes on individual neuropsychological tasks from sham-controlled RCTs where rTMS was administered to the DLPFC in depressed patients has recently been published [46]. No significant effect size for improvements with active compared to sham rTMS treatment was found.

For the purpose of this review, it is of interest that some studies used more specific tests to assess attentional processes. In a double-blind, placebo-controlled, crossover, within subjects design study, sixteen depressed patients performed a modified task switching paradigm, before and after receiving HF rTMS versus placebo rTMS over the left DLPFC [41]. One session of HF rTMS over the left DLPFC had a specific beneficial effect on task-switching performance, whereas mood remained stable. The same research group also found that after 2 weeks of HF rTMS over the left DLPFC, depressive symptoms improved in more than half of a therapy-resistant population [42]. After a single session, mood did not improve but attentional control was increased solely within the group of treatment responders. Of course, it needs to be considered that depression has very broad negative effects on cognitive function, so that a relieving of depressive symptoms might in turn have overall positive effects on cognition.

A more recent study examined whether acute and long-term HF deep rTMS to the DLPFC can attenuate attentional deficits associated with Major Depressive Disorder [47]. Twenty-one patients and 26 matched control subjects were characterized with the Beck Depression Inventory and the Sustained Attention to Response Task (SART) at baseline. Patients were retested following a single session and after 4 weeks of HF (20 Hz) deep rTMS applied to the DLPFC. To control for the practice effect, the controls were reassessed with the SART two further times. The patients exhibited deficits in sustained attention and cognitive inhibition. Both acute and long-term HF frontal repetitive dTMS ameliorated sustained attention deficits in the patient group. Improvement after acute dTMS was related to attentional recovery after long-term dTMS. It should be noted that longer-term improvement in sustained attention was not related to antidepressant effects of dTMS treatment.

Kavanaugh et al. examined recently the neurocognitive results of a randomized, double-blind, sham-controlled trial with an investigational 2-coil rTMS device [47]. The authors included patients with antidepressant treatment or treatment-intolerant major depressive disorder. A significant effect of active rTMS was observed for the quality of episodic memory, while there were no effects for continuity and power of attention as well as for working memory.

### 5.2. Schizophrenia

In patients with chronic schizophrenia, no significant change of cognitive performances, including the d2 attention task to assess attentional capacity, was observed as the result of a HF rTMS treatment [43]. Wölwer and coworkers also failed to find any significant cognitive effects in patients with schizophrenia who received HF rTMS [50].

In another study with schizophrenia patients, excitatory rTMS applied to the DLPFC was found to improve, among other cognitive functions, the selective and divided attention, as assessed by means of the Tübinger Aufmerksamkeitsprüfung [51].

Active rTMS can lead to a statistically significant reduction on the Scale for the Assessment of Negative Symptoms total score and of all domains of negative symptoms of schizophrenia, including impaired attention [52].

### 5.3. Autism Spectrum Disorder

Event-related brain potentials (ERPs) provide high temporal resolution measures of neuronal activity associated with several perceptual and cognitive processes. Sokhadze et al. assessed post-TMS differences in 13 subjects with autism [53]. The authors examined amplitude and latency of early and late attention-orienting frontal ERP components, indicating improved attentional processing. After rTMS, the parieto-occipital P50 amplitude decreased to novel distractors but not to targets; also, the amplitude and latency to targets increased for the frontal P50 while decreasing to nontarget stimuli.

Twenty-five subjects with autism spectrum disorder (ASD) were assessed in order to characterize selective attention using illusory figures before and after 12 sessions of rTMS applied bilaterally to the DLPFC [54]. This study was conducted in a controlled design where a waiting-list of 20 children with autism spectrum disorder was examined with the same time-interval, but with no rTMS intervention. A significant increase in amplitude of both N200 and P300 components as well as a significant reduction in response errors as a result of rTMS were detected.

The same research group also found, in 124 high functioning ASD children, that 18 sessions of rTMS applied over the DLPFC facilitates cognitive control, attention, and target stimuli recognition by improving discrimination between task-relevant and task-irrelevant illusory figures in an oddball test [55].

### 5.4. Attention Deficit Hyperactivity Disorder

In a crossover double-blind randomized, sham-controlled pilot study, patients with ADHD received either a single session of HF rTMS directed to the right DLPFC (real rTMS) or a single session of sham rTMS [56]. The post-real rTMS attention score improved significantly compared to the prereal rTMS attention score. rTMS had no effect on measures of mood and anxiety, and sham rTMS showed no effects.

In a more recent study, twenty daily sessions were conducted in patients diagnosed as having ADHD, using the bilateral HF dTMS coil in order to stimulate the PFC. The Conners’ Adult ADHD Rating Scale questionnaire and a computerized continuous performance test, the Test of Variables of Attention, were used for the assessment of cognitive functions. No differences in clinical outcomes were observed between groups receiving real dTMS or sham TMS [57].

### 5.5. Addiction

HF (10 Hz) rTMS of the left DLPFC was found to improve emotional attention of 31 methamphetamine addicts [58]. The attention bias effect to negative information persisted in the active rTMS group over two weeks.

An fMRI study in 26 recently detoxified alcohol-dependent patients documented effects of accelerated HF rTMS applied to the right DLPFC [59]. The findings suggest that the intervention did not manifestly affect the craving neurocircuit during an alcohol-related cue-exposure, but instead it may have influenced the attentional network. In fact, brain activation changes after one and 15 HF rTMS sessions were observed in regions associated with the extended reward system and the default mode network, respectively, during the presentation of event-related alcohol cue-reactivity paradigms.

### 5.6. Alzheimer’s Disease

A single study has examined the effects of HF rTMS, applied over the DLPFC on behavioral and psychological symptoms of dementia as well as on cognitive function in 52 patients with Alzheimer’s disease (AD) [60].

The intervention group, which was treated with 20 Hz rTMS five days a week for four weeks, showed significantly lower scores (i.e., greater improvement) than the control group on the Alzheimer’s Disease Assessment Scale-Cognitive (ADAS-Cog) total score, as well as on all four ADAS-Cog factor scores (memory, language, constructional praxis, and attention).

## 6. Discussion

This review highlights that rTMS applied over the DLPFC can positively influence the attentional function in subjects with several psychiatric disorders. The outcome measures were not uniform but mostly dealt with attentional performance.

Some studies revealed that prefrontal rTMS could exert procognitive effects on executive function and attention in patients with depression [3]. Antidepressant effects of rTMS could be related to the same neurochemical mechanisms that underlie cognitive functioning, or just facilitate the normal cognitive function that was repressed because of the severe effects depression has on overall physical and cognitive functioning. It has been hypothesized that the extent of antidepressant effects could be considered as second-order long-term effects possibly related to primary alternations in cognitive functioning. Concurrence of depression and cognitive dysfunctions is well known in a wide range of clinical populations [61]. In particular, impaired cognition is closely related to depressive symptoms in AD [62,63], thus possibly potentiating the devastating effects of the disease itself or being an early sign of neural dysfunction [64].

In patients with schizophrenia, imaging studies have demonstrated abnormalities in the left globus pallidus, which lead to widespread hypometabolism affecting the frontal lobes, especially the DLPFC and the anterior cingulate gyrus [65]. Furthermore, abnormalities of visually orienting the frontal lobes/executive attentional network could interact with the parietal lobes/orienting network to affect the initiation of attentional shift, thus leading to abnormalities of visual orienting [66]. It is therefore of interest that rTMS to the DLPFC could improve attentional functioning in this patient population [38]. However, the findings were contradictory, as other studies could not identify any beneficial effects. A more systematic investigation comparing the different parameters of TMS to each other may shed more light on the mechanisms of action.

The results of some studies support the use of LF rTMS as a modulatory tool to alter the disrupted balance between cortical excitation and inhibition in autism. LF rTMS application to DLPFC would result in an alteration of the abnormal excitatory/inhibitory ratio through the activation of inhibitory GABAergic double bouquet interneurons.

Similarly, in patients suffering from ADHD initial findings suggest the possibility that attentional difficulties can be improved by using HF rTMS applied to the right DLPFC, and have encouraged future research [41]. However, the evidence from a more recent study does not support the effectiveness of bilateral prefrontal stimulation to treat adult ADHD [42]. Due to the small sample size, these preliminary results should be interpreted with caution.

rTMS can significantly improve, among other cognitive functions, attentional impairment that often accompanies AD. Impairments in visual attention and visual information processing have been identified as part of the neuropsychological features of AD, even in its earliest stages, and dissociations in visual attention deficits have been detected also in mild cognitive impairment (MCI) using a measure that assesses simple, divided, and selective attention [67]. It is unclear whether the memory impairment in patients with amnestic MCI (aMCI) and AD is associated with attentional deficits. An fMRI study revealed that there are changes in the functional network subserving divided attention in patients with aMCI, as reflected in the attenuated activation of PFC [68]. Interestingly, depressive symptoms in AD patients increase the deficits of cognitive flexibility and divided attention [69].

This review has some limitations. First of all, there is considerable variability between studies in patients with different neuropsychiatric diseases. Very few trials have used exactly the same study design. The stimulation protocols, with respect to frequency, intensity, orientation of the coil, pattern, number of pulses by train, total number of pulses, duration of stimulation, frequency and intensity of stimulation, number of sessions delivered, are highly heterogeneous. Therefore, estimating the real effectiveness and reproducibility is very difficult. Systematic investigation of the effects of the various stimulation protocols are highly warranted, because the border between effectiveness and ineffectiveness may be very small and occurs somewhere in the dimensions spanned by the abovementioned parameters.

Furthermore, we have included in this review only studies employing specific cognitive tests/tasks focusing on attention, even if working memory and other executive functions are strongly correlated with this cognitive domain. Indeed, the role of the right DLPFC and of the right posterior parietal cortex (PPC) in controlling the interaction between working memory and attention during a visual search has been explored using rTMS in a recent study [70]. Both the rDLPFC and the right PPC were found to be critical for controlling working memory biases in human visual attention. However, the broader scope of including executive functions should be addressed in another systematic summarizing work; possibly a meta-analysis could be conducted given that the study protocols were more comparable.

It should be considered that most therapeutic attempts are based on rTMS techniques aiming at enhancing cortical excitability, in particular HF rTMS. However, the underlying pathophysiologic mechanisms differ among the various neurological and psychiatric diseases which can be treated with this noninvasive brain stimulation technique. Therefore, appropriate testing of cortical physiology before and after therapeutic interventions is needed.

## 7. Conclusions

In conclusion, a better understanding of attention networks could allow targeting the most suitable area of the brain according to the specific attention domain affected. Moreover, a detailed examination of the best stimulation frequency, surface or deep stimulation, duration and intensity of the intervention, among other important core features of TMS-protocols, should be done when moving closer to clinical application of TMS to treat attentional deficits.

Despite the above-mentioned limitations, this review indicates that neuromodulatory techniques such as rTMS are promising approaches to be used as attentional enhancers in people with neuropsychiatric conditions where impaired attention is a prominent feature.

## Figures and Tables

**Figure 1 jcm-08-00416-f001:**
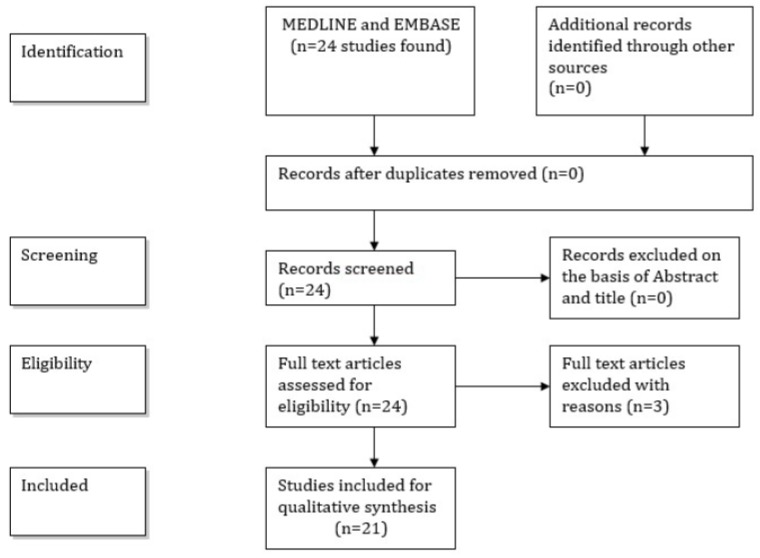
Flow-chart showing the selection/inclusion process.

**Table 1 jcm-08-00416-t001:** Demographic characteristics of the patients in the included studies.

Studies	No	Gender	Mean Age	Disease Duration	Education
		M/F	(y)	(y)	(y)
**Depression**					
Speer et al., 2001 [28]	18	-	45 ± 7	-	-
Höppner et al., 2003 [29]	30	8/22	56.4 ± 11.1	-	-
Januel et al., 2006 [30]	27	6/21	37.78 ± 11.27	77.77 ± 90.82 mo	-
Levkovitz et al., 2009 [31]	23 H1	12/11	45.57 ± 13.34	13.96 ± 2.96	-
	22 H2	11/11	45.77 ± 11.99	13.00 ± 2.12	-
	11 HIL 110%	3/5	44.27 ± 11.36	15.45 ± 2.02	-
	8 HIL 120%	10/10	49.88 ± 9.52	13.13 ± 2.81	-
Vanderhasselt et al., 2009a [32]	16	6/10	42 ± 11.2		
Vanderhasselt et al., 2009b [33]	15	6/9	45.6 ± 5.87	-	-
Ullrich et al., 2012 [34]	Active 22	31.8/68.2%	56.9/10.2%	-	-
	Sham 21	42.9/57.1%	54.1/7.8%		
Naim-Feil et al., 2016 [35]	21	10/11	44 ± 9	15 ± 3	-
Kavanaugh et al., 2018 [36]	Active 43	10/31	45.84 ± 11.87	17.94 ± 3.7	-
	Sham 41	12/31	47.95 ± 12.78	15.59 ± 9.17	
**Schizophrenia**					
Mittrach et al., 2010 [37]	Active 18	14/4	34.5 ± 0.5	5.7 ± 5.2	-
	Sham 14	11/3	34.4 ± 10.5	5.6 ± 8.7	
Guse et al., 2013 [38]	Active 13	10/3	37 (22-58)	15.5	-
	Sham 12	9/3	36 (20-51	12.6	
Prikryl et al., 2013 [39]	Active 23	23/0	31.6 ± 8.04	4.91 ± 5.09 y	12.43 ± 2.06 y
	Control 17	17/0	33.94 ± 9.98	5.89 ± 7.91 y	12.44 ± 1.97
Wölwer et al., 2014 [40]	Active 18	14/4	34.3 ± 5.7	5.7 ± 5.2	-
	Sham 14	11/3	34.4 ± 5.6	5.6 ± 8.7	
**Attention deficit hyperactivity disorder**					
Bloch et al., 2010 [41]	13	7/6	-	-	-
	Active 9	6/3	32 ± 11		
Paz. et al., 2017 [42]	Sham 13	8/5	30.85 ± 6.82	-	-
**Alzheimer disease**					
Wu et al., 2015 [43]	Active 26	10/16	71.4 ± 4.9	5.1 ± 1.5	11.4 ± 2.7 y
	Control 26	11/15	71.9 ± 4.8	5.1 ± 1.5	11.5 ± 2.1 y
**Autism**					
Sokhadze et al., 2010 [44]	13	12/1	15.6 ± 5.8	-	-
Casanova et al., 2012 [45]	45	39/6	13 ± 2.7	-	-
Sokhadze et al., 2018 [46]	112	93/19	13.1 ± 1.78	-	-
**Addiction**					
Herremans et al, 2015 [47]	26	17/9	45.2 ± 9.3	-	-
Zang et al., 2018 [48]	31	31/0	43 ± 9.15	13 ± 7.45	-

no. = number of patients; M = male; F = female; y = years; mo. = months, “-“ not reported.

**Table 2 jcm-08-00416-t002:** Description of the repetitive transcranial magnetic stimulation (rTMS) interventions in the included studies.

Studies	Stimulation Parameters					Outcome Measures	Principal Findings
	Position	Intensitity	Frequency	Total Pulses Per Session	No. Sessions		
**Depression**							
Speer et al., 2001 [28]	L DLPFC	100% MT	20 Hz 1 Hz	1600	10	Continuous Performance Task	No significant changes
Hoeppner et al., 2003 [29]	L DLPFC R DLPFC	80% MT	20 Hz 1 Hz	?	10	d2 Test	No significant changes
Januel et al., 2006 [30]	R DLPFC	90% MT	1 Hz	?	16	Auditory and visual attention span	No significant differences
Levkovitz et al., 2009 [31]	H-Coil DLPFC	120% MT	20 Hz	1689	20	CANTAB, RVP	↑ RVP performances
Vanderhasselt et al., 2009a [32]	L DLPFC	110% MT	10 Hz	1560	10	VAS Self-paced switching task	↑ Attentional processes
Vanderhasselt et al., 2009b [33]	L DLPFC	110% MT	10 Hz	1560	10	Self-paced switching task	↑ Attentional control
Ullrich et al.; 2012 [34]	L DLPFC	110% MT	30 Hz 1 Hz	1800 990	15	ZVT, SKT	↑ Processing speed performance ↑
Naim-Feil et al., 2016 [35]	H-Coil L > R DLPFC	120% MT	20 Hz	1680	1 (*n* = 21) 20 (*n* = 13)	BDI, SART	↓ Sustained attention deficits
Kavanaugh et al., 2018 [36]	2-coil L > R DLPF	120% MT	10 Hz	3000	20	CDR System	↑ Continuity and power of attention
**Schizophrenia**							
Mittrach et al., 2010 [37]	L DLPFC	110% MT	10 Hz	1000	10	d2 Test	No significant changes
Guse et al., 2013 [38]	L DLPFC	110% MT	10 Hz	1000	15	TAP	Significant time-by-stimulation interaction in divided attention
Prikryl et al., 2013 [39]	L DLPFC	110% MT	10 Hz	2000	15	SANS	↓ SANS total score + all domains of negative symptoms
Woelwer et al., 2014 [40]	L DLPFC	110% MT	10 Hz	10000	10	d2 Test	No significant changes
**Attention deficit hyperactivity disorder**							
Bloch et al., 2010 [41]	R DLPFC	100% MT	20 Hz	?	?	PANAS, VAS attention/mood CANTAB	↑ VAS for attention
Paz. et al., 2017 [42]	H-Coil L/R DLPFC	120% MT	18 Hz	1980	20	TOVA, CAARS	No differences sham/active rTMS
**Alzheimer disease**							
Wu et al., 2015 [43]	L DLPFC	80% RMT	20 Hz	1200	20	BEHAVE-AD, ADAS-Cog scores	Improvement in all ADAS-Cog scores
**Autism**							
Sokhadze et al., 2010 [44]	L DLPFC	90 % RMT	0,5 Hz	150	6	ABC, SCR, RBS Early and late ERP components	Improvement of error percentage to targets P50 parieto-occipital↓, frontal ↑
Casanova et al., 2012 [45]	L/R DLPFC	90 % RMT	≤ 1 Hz	150	12	Selective attention illusory figures ERP indices of selective attention	↓ in response errors ↑ N200 and P300 components
Sokhadze et al., 2018 [46]	L/R DLPFC	90 % RMT	1 Hz	180	18	Visual oddball with Kanizsa figures Stimulus and response-locked ERP	↑ Motor responses accuracy ↑ Early and later-stage ERP indices
**Addiction**							
Herremans et al, 2015 [47]	R DLPFC	110 % RMT	20 Hz	1560	15	AUQ, OCDS	Cue-induced alcohol craving was not altered
Zang et al., 2018 [48]	L DLPFC	90 % RMT	10 Hz	2000	14	Chinese Affective Picture System	Improvement of emotional attention in meth addicts

Table legend: R = right; L = left; DLPFC = dorsolateral prefrontal cortex; MT = motor threshold; CANTAB = Cambridge Neuropsychological Test Automated Battery; RVP = Rapid Visual Processing; VAS = Visual Analogue Scale; ZVT = Zahlen-Verbindungs-Test; SKT = Syndrom-Kurztest; BDI = Beck depression Inventory; SART = Sustained Attention to response task; CDR System = Cognitive Drug Research Computerized Assessment System; TAP = Test of Attentional Performance; SANS = Scale for the Assessment of Negative Symptoms; PANAS = Positive and Negative Affect Schedule; TOVA = Test of Variables of Attention; CAARS = Conners’ Adult ADAH Rating Scale; BEHAVE-AD = Behavioral Pathology in Alzheimer’s Rating Scale; ADAS-Cog = Alzheimer’s Disease Assessment Scale-Cognitive; ABC = Aberrant Behavior Checklist; SCR = Social Responsiveness Scale; RBS = Repetitive Behavior Scale; AUQ = Alcohol Urge Questionnaire; OCDS = Obsessive Compulsive Drinking Scale; ERP = event-related potentials; ↑ = enhancement; ↓ = reduction.

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
