# Peer review of "Effects of Repetitive Transcranial Magnetic Stimulation over Prefrontal Cortex on Attention in Psychiatric Disorders: A Systematic Review"

_jcm, 2019, doi:10.3390/jcm8040416_

Reviewer 1 Report

Review of manuscript by  Hauer et al. “Effects of repetitive transcranial magnetic stimulation over prefrontal cortex on attention in psychiatric disorders: a systematic review” submitted to Journal of Clinical Medicine.

 The paper reviews 21 papers eligible for the systematic analysis using selected criteria to investigate outcomes of repetitive transcranial magnetic stimulation (rTMS)  in several psychiatric disorders (depression, schizophrenia, ADHD, Alzheimer disease, autism, and some of substance use disorder, mostly alcohol and methamphetamine dependence).  This is a project that targets analysis of usefulness of treatment psychiatric and/or neurological disorders presenting with executive function deficits as post-TMS improvement are expected to confirm enhancement of attention by neuromodulation. The goal of improving attention using TMS is based on an acceptable scientific premise as certain TMS protocols are known to improve frontal functions in various psychopathologies.  Therefore, the topic as such analysis of rTMS effects on attention in psychiatric disorders is interesting and timely, however, there are some concerns that somehow slightly lower enthusiasm to this particular review. The method of rTMS is used well over the couple of decades and definitely more studies could be selected. Discussion is sufficiently well detailed in regards to specific selected psychopathologies, though probably some (addiction) should not be included as only 2 studies and different substance users were reviewed. For the general introduction to the area of magnetic neuromodulation on attention this mini-review could be considered as good enough, though it can hardly be called a systematic review. For an entry level non-specialist journal audience (as compared to brain stimulation and attention research specialist) this manuscript might be of interest and important to attract more attention to potentials of this very promising neuromodulation technique.

There are, nevertheless, a whole list of technicality issues, most of them probably just very minor, but definitely needing revision and corrections. Below are listed clear typos/errors  in the Table 1. In several occasions data for columns of  “Disease duration” and “Education”  are messed up, placed in the wrong columns. It was hard to believe that patients with depression, Alzheimer could possibly have only 5-to-6 years of education and disorders duration 12 years. Check of original literature revealed that in Prikryl et al., 2013,  Januel et al., 2006, and Wu et al. 2015 data in these two columns of the table were placed in the wrong place. Several other similar incorrect data was found in this table, for example in Autism section in Sokhadze et al. studies it seemed not logical to do first 112  subjects and in 2018 only 13.  Check with original paper info showed that years were mixed , first in the section are data for Sokhadze et al., 2018, while the third one should be Sokhadze et al., 2010.  Probably this Table 1 needs very careful revision and double-check for correctness of data presented there. Also it is not clear why table is in image (scanned or PDF) format. The tables are prepared in a very poor quality, should be prepared more carefully for such type of mini review with only limited number of papers.

In addition, the text has numeration of references 1 through 70, not in alphabetic order and this makes finding references cited in table studies difficult for the reader. It would make sense include as well citation number in both tables (for example Wu et al., 2015 [49]). There were noted as well some technicality issues, again manor ones in the References section. References: For [34] and [35]  should be added a and b letters (as they are in the table but not in References). Some references (in particular 36,37, 43,44, 48, 63, 64 and 70) have capital letters in titles of the papers, needs to be unified in style with other citations.

Author Response

The paper reviews 21 papers eligible for the systematic analysis using selected criteria to investigate outcomes of repetitive transcranial magnetic stimulation (rTMS)  in several psychiatric disorders (depression, schizophrenia, ADHD, Alzheimer disease, autism, and some of substance use disorder, mostly alcohol and methamphetamine dependence).  This is a project that targets analysis of usefulness of treatment psychiatric

Response to Reviewer 1 comments:

The paper reviews 21 papers eligible for the systematic analysis using selected criteria to investigate outcomes of repetitive transcranial magnetic stimulation (rTMS)  in several psychiatric disorders (depression, schizophrenia, ADHD, Alzheimer disease, autism, and some of substance use disorder, mostly alcohol and methamphetamine dependence).  This is a project that targets analysis of usefulness of treatment psychiatric and/or neurological disorders presenting with executive function deficits as post-TMS improvement are expected to confirm enhancement of attention by neuromodulation. The goal of improving attention using TMS is based on an acceptable scientific premise as certain TMS protocols are known to improve frontal functions in various psychopathologies.  Therefore, the topic as such analysis of rTMS effects on attention in psychiatric disorders is interesting and timely, however, there are some concerns that somehow slightly lower enthusiasm to this particular review. The method of rTMS is used well over the couple of decades and definitely more studies could be selected. Discussion is sufficiently well detailed in regards to specific selected psychopathologies, though probably some (addiction) should not be included as only 2 studies and different substance users were reviewed. For the general introduction to the area of magnetic neuromodulation on attention this mini-review could be considered as good enough, though it can hardly be called a systematic review. For an entry level non-specialist journal audience (as compared to brain stimulation and attention research specialist) this manuscript might be of interest and important to attract more attention to potentials of this very promising neuromodulation technique.

There are, nevertheless, a whole list of technicality issues, most of them probably just very minor, but definitely needing revision and corrections. Below are listed clear typos/errors  in the Table 1. In several occasions data for columns of  “Disease duration” and “Education”  are messed up, placed in the wrong columns. It was hard to believe that patients with depression, Alzheimer could possibly have only 5-to-6 years of education and disorders duration 12 years. Check of original literature revealed that in Prikryl et al., 2013,  Januel et al., 2006, and Wu et al. 2015 data in these two columns of the table were placed in the wrong place. Several other similar incorrect data was found in this table, for example in Autism section in Sokhadze et al. studies it seemed not logical to do first 112  subjects and in 2018 only 13.  Check with original paper info showed that years were mixed , first in the section are data for Sokhadze et al., 2018, while the third one should be Sokhadze et al., 2010.  Probably this Table 1 needs very careful revision and double-check for correctness of data presented there. Also it is not clear why table is in image (scanned or PDF) format. The tables are prepared in a very poor quality, should be prepared more carefully for such type of mini review with only limited number of papers.

Response: We firstly thank for the favorable comments and also apologize for the mistakes in the tables.

We have prepared the Tables more carefully, and the above mentioned errors have been corrected.

In addition, the text has numeration of references 1 through 70, not in alphabetic order and this makes finding references cited in table studies difficult for the reader. It would make sense include as well citation number in both tables (for example Wu et al., 2015 [49]). There were noted as well some technicality issues, again manor ones in the References section. References: For [34] and [35]  should be added a and b letters (as they are in the table but not in References). Some references (in particular 36,37, 43,44, 48, 63, 64 and 70) have capital letters in titles of the papers, needs to be unified in style with other citations.

Response: We updated the manuscript according to the suggestions and included citation number in both tables.

For references [34] and [35] we added “a” and “b”. In addition, we unified the style of the references.

Reviewer 2 Report

The authors provide a review of the effect of rTMS administered to the dorsolateral prefrontal cortex on attention. This review is useful to provide an update in a field which is continuing to gain traction. In particular, focusing on a specific cognitive domain (i.e. attention) across multiple diseases is helpful, and in line with RDOC efforts.

-          Abstract: “…has the potential to target core features of ASD” – because of evidence from the 3 articles which looked at individuals with ASD, or from the studies looking at depression and schizophrenia?

-      Abstract: Alzheimer's disease is often categorized as a neurological disease, not psychiatric

-      The review states it will look at rTMS in psychiatric diseases, but there is also a section on healthy population. This section is helpful, just should be alluded to in the abstract/intro. It would actually be useful to move the healthy population section to the beginning of the review, to provide the reader with information about a 'baseline' to compare to

-          Line 92: Unfamiliar with the level ranking – either define the system or remove

-          The authors devote text and a figure to explain the selection criteria for included papers, but fail to indicate the reason 3 papers were excluded (other than stating “with reasons”) – what are these reasons?

-          In cases where the total pulses per session are not known, is the duration of stimulation known?

-          Important to state if outcome measure was tested concurrently with stimulation, or following stimulation; should also state how change in performance was determined (e.g. increase in performance of task during stimulation relative to sham stimulation; relative to TMS with different stimulation parameters)

-          Would be helpful to include reference numbers in the table for studies (to allow easy comparison to descriptions of some studies in the text)

-          Lines 205-207: Specify that these changes indicate improve attentional processing

-          A summary sentence at the end of each disease category would be helpful

-          Lines 273 – 275: Different types of attention and which tasks assess them is an important point; please expand on this. If possible, categorize tasks when describing them previously in the review into major subheadings.

Author Response

Response to Reviewer 2 comments

The authors provide a review of the effect of rTMS administered to the dorsolateral prefrontal cortex on attention. This review is useful to provide an update in a field which is continuing to gain traction. In particular, focusing on a specific cognitive domain (i.e. attention) across multiple diseases is helpful, and in line with RDOC efforts.

-          Abstract: “…has the potential to target core features of ASD” – because of evidence from the 3 articles which looked at individuals with ASD, or from the studies looking at depression and schizophrenia?

Response: This statement was made on the basis of the three articles which studied individuals with ASD.

-      Abstract: Alzheimer's disease is often categorized as a neurological disease, not psychiatric

Response: Neuropsychiatric symptoms (NPS) are core features of Alzheimer’s disease and related dementias (10.1016/j.jalz.2011.05.2410).

-      The review states it will look at rTMS in psychiatric diseases, but there is also a section on healthy population. This section is helpful, just should be alluded to in the abstract/intro. It would actually be useful to move the healthy population section to the beginning of the review, to provide the reader with information about a 'baseline' to compare to

Response: We added following sentence: We also reviewed and discussed the studies assessing the effects of rTMS on attention in the healthy population.

-          Line 92: Unfamiliar with the level ranking – either define the system or remove

Response: We have clarified in the revised version by adding: “There is sufficient body of evidence to accept with level of recommendation A (definite efficacy, Evidence Based Health Care)”

-          The authors devote text and a figure to explain the selection criteria for included papers, but fail to indicate the reason 3 papers were excluded (other than stating “with reasons”) – what are these reasons?

Response: These three studies failed to meet the inclusion criteria after study of the full text, which is reported in lane 119.

-          In cases where the total pulses per session are not known, is the duration of stimulation known?

Response: The reports of duration were less consistent than frequency and total pulses. Thus we did not report.

-          Important to state if outcome measure was tested concurrently with stimulation, or following stimulation; should also state how change in performance was determined (e.g. increase in performance of task during stimulation relative to sham stimulation; relative to TMS with different stimulation parameters)

Response: The outcome measures were not uniform but mostly dealt with attentional performance. We added this information to the first paragraph of the discussion, the outcome measures are also noted in the coverage of the individual studies.

-          Would be helpful to include reference numbers in the table for studies (to allow easy comparison to descriptions of some studies in the text):

Response: Included as suggested.

-          Lines 205-207: Specify that these changes indicate improve attentional processing.

Response: We specified that these changes indicate improved attentional processing.

-          A summary sentence at the end of each disease category would be helpful.

Response: We believe that there is redundancy since summaries are provided in the discussion.

-          Lines 273 – 275: Different types of attention and which tasks assess them is an important point; please expand on this. If possible, categorize tasks when describing them previously in the review into major subheadings.

Response: We believe that such and expansion of this point get beyond the aims of this study. However, if necessary, we will this expand and tasks categorize.

Round  2

Reviewer 1 Report

The authors responded to all critical comments and suggestions that I had regarding initial version of this manuscript. I did not see formal response to reviewers but changes that i suggested were accepted by the authors. I do not have any objections.